# Surgical Management of Groove Pancreatitis: A Case Report

**DOI:** 10.3390/jpm13040644

**Published:** 2023-04-08

**Authors:** Aristeidis Ioannidis, Alexandra Menni, Georgios Tzikos, Eleni Ioannidou, Georgia Makri, Angeliki Vouchara, Patroklos Goulas, Eleni Karlafti, Elizabeth Psoma, Xanthipi Mavropoulou, Daniel Paramythiotis

**Affiliations:** 11st Propaedeutic Department of Surgery, Aristotle’s University of Thessaloniki, AHEPA University Hospital, 54634 Thessaloniki, Greece; 2Emergency Department, Aristotle’s University of Thessaloniki, AHEPA University Hospital, 54634 Thessaloniki, Greece; 3Department of Radiology, Aristotle’s University of Thessaloniki, AHEPA University Hospital, 54634 Thessaloniki, Greece

**Keywords:** groove pancreatitis, alcohol, case report, duodenal stenosis, surgical management

## Abstract

Groove pancreatitis (GP) is a chronic type of pancreatitis involving the groove area between the head of the pancreas, the duodenum, and the common bile duct. Alcohol abuse is one of the main pathogenetic factors, although its etiology is not clearly defined. Differential diagnosis of pancreatic disorders remains difficult. The lack of diagnostic management and the restrictive number of patients are the main barriers. This article presents a case of a 37-year-old male diagnosed with GP after several episodes of epigastric pain and vomiting, with a history of chronic alcohol consumption. The patient’s radiological and laboratory results excluded the possibility of malignancy and suggested the diagnosis of groove pancreatitis with duodenal stenosis. After initial conservative treatment failed, surgical management was decided. A gastroenteroanastomosis was made in order to bypass the duodenum aiming for a total resolution of the symptoms and an uneventful recovery of the patient. Although most studies suggest pancreatoduodenectomy (Whipple’s procedure) as the treatment of choice, a less major procedure can be performed in evidence of malignancy absence.

## 1. Introduction

Groove pancreatitis (GP), also known as paraduodenal pancreatitis, is a form of chronic disease located in the pancreaticoduodenal groove, defined by the head of the pancreas, the duodenum, and the common bile duct. It is categorized into two forms: pure and segmental. Chronic alcohol consumption and heavy smoking seem to have a strong connection with GP’s pathogenesis, which is not clearly determined yet [1]. GP usually affects 40–50-year-old alcoholic male patients who are presented with epigastric pain, vomiting, and weight loss. Diagnosing groove pancreatitis can be extremely challenging since it imitates pancreatic cancer. Typical radiological findings are cystic lesions, duodenal wall thickening, and poorly enhanced mass between the duodenal wall and the head of the pancreas. The treatment options include conservative, endoscopic, and surgical interventions, such as pancreatoduodenectomy and duodenal bypass. This article presents a case of groove pancreatitis with surgical management [2].

## 2. Case Presentation

A 37-year-old male patient presented to the emergency department complaining of postprandial epigastric pain radiating to the back and vomiting episodes relating to food intake over the last fourteen days. Furthermore, he reports recent weight loss and weakness because of his inability to feed due to the severe symptoms mentioned above. The patient has a history of chronic alcohol consumption and heavy smoking over the past 10 years, and it was the only one mentioned from the patient’s medical history. The physical examination showed tenderness in the upper abdomen and the right hypochondric region. According to his laboratory exams, serum amylase was within the normal range, and the level of lipase was 90 U/L (normal levels: 13–60 U/L). Tumor markers, CA 19.9, and carcinoembryonic antigen (CEA) were found normal. An abdominal Computed Tomography (CT) was performed. It revealed a thickening of the pyloric and the duodenal wall and a 26 mm hypodense mass at the level of the pancreatic head visible in the arterial phase. An upper GI endoscopy was then conducted, which showed luminal stenosis, edema, and redness of the duodenum. In a second CT performed some days later, the pancreatic head appeared enlarged and with fuzzy limits. In addition, cystic lesions (maximum diameter of 12 mm) and fibrous tissue were found in the head of the pancreas near the duodenal wall (Figure 1). Based on those clinical and radiological findings, the suspected diagnosis was groove pancreatitis or posterior ulcer of the duodenal bulb, and a conservative approach was decided initially. Antibiotics, analgesics, proton pump inhibitors, and also fluids, and parental nutrition were administered in combination with pancreatin. The patient’s clinical and laboratory status was significantly improved, and he was discharged from the hospital a few days later, and strict abstinence from alcohol was recommended. However, the patient relapsed and was readmitted to the hospital due to abdominal pain and vomiting. A new CT was conducted, which confirmed the previous findings and additionally showed enlargement of the three known cystic lesions and the presence of ectopic pancreatic tissue in the posterior wall of the pyloric antrum. A second upper GI endoscopy was followed, which revealed inflammation, edema, and redness of the duodenal wall, causing the luminal obstruction. In need of further investigation, an MRI of the upper abdomen was performed (Figure 2), which confirmed the CT findings with the presence of a new cystic lesion between the left hepatic lobe and the pylorus, with all the cystic lesions being classified as necrotic ones. The patient showed no signs of improvement with the conservative means of treatment: weight loss continued while the inability to consume food orally became more intense. Hence, the surgical management of the patient was decided. Since no malignancy was suspected, the surgical procedure of choice was gastroenteroanastomosis, aiming to bypass the duodenal obstruction and relief the patient from the symptoms of dysphagia or vomiting. The surgery consisted of an anastomosis between the posterior wall of the stomach and the jejunum (gastrojejunal anastomosis) through the transverse mesocolon and an additional Braun anastomosis (jejunojejunal anastomosis) to prevent the development of postoperative alkaline gastritis. The patient’s recovery was uneventful, and he was discharged from the hospital on the 10th postoperative day, with no fever, hemodynamically and gasometrically stable, after adequate feeding without any disturbances.

## 3. Discussion

Groove pancreatitis (GP), also known as paraduodenal pancreatitis, is a form of chronic pancreatitis located in the pancreaticoduodenal groove [3]. The groove area refers to the anatomic space between the pancreatic head medially, the second segment of the duodenum laterally, the third segment of the duodenum and inferior vena cava posteriorly, and the first segment of the duodenum superiorly [4]. GP is mainly associated with long-term alcohol intake and has also been reported in patients with known peptic ulcer disease and among children with a congenital origin [5].

In 1973, Becker was the first to describe a type of chronic focal pancreatitis located in the pancreatoduodenal area, with the German name of “Rinnen pancreatitis” [3], although, in 1982, Stolte et al. introduced the term “groove pancreatitis” [1]. In 1991, Becker and Mischke recognized two forms of GP: the pure form, which is characterized by inflammatory cells with fibrous lesions, and affects the area of the pancreatic head and duodenum exclusively while the pancreatic parenchyma remains intact [6], and the segmental form, in which the scar tissue affects not only the groove but also the pancreatic head and body near the duodenal wall, where, in addition to the pancreas head, the surrounding structures, such as the common bile duct and duodenum, are affected, causing stenosis of the pancreatic duct, leading to symptoms such as jaundice, abdominal pain, postprandial vomiting or weight loss due to poor nutrition [1,3].

The pathogenesis of GP is not yet precisely determined, although there seems to be a strong correlation between chronic alcohol consumption and heavy smoking [7]. More precisely, alcohol consumption causes a decrease in bicarbonate secretion by increasing the viscosity of the pancreatic fluid, thus causing stagnation in pancreatic secretion within the pancreatic ducts. In this way, the pressure within the Santorini duct increases, ultimately forming pseudocysts [8]. Brunner gland hyperplasia, ectopic pancreatic tissue, true duodenal wall cysts, history of gastrectomy, gastroduodenal ulcer, biliary diseases, and the presence of anatomic abnormalities are also associated with the disease [9,10].

Patients diagnosed with GP are mostly alcoholic men in their fourth or fifth decade of life [7]. The clinical presentation is similar to that of chronic pancreatitis and includes postprandial epigastric pain radiating to the back and vomiting caused by duodenal stenosis, with significant weight loss [9,11]. Jaundice may also occur in some cases [12], as well as diabetes mellitus [13]. Laboratory results reveal a concomitant twofold or threefold increase of amylase and lipase concentration in the serum at the proximal of 80% of the patients, while tumor markers, such as CA19-9 and carcinoembryonic antigen-CEA, are usually within the normal limits [14,15,16].

Regarding the diagnostic imaging findings of GP, several methods have been used through the years, although their diagnostic management remains challenging. Contrast-enhanced abdominal computed tomography (CT) shows a hypodense poorly enhanced lesion between the pancreatic head and the duodenal wall. Cysts can be found in the groove area or/and the duodenal wall, which is also thickened, causing duodenal stenosis [9,17,18]. The early phase of a contrast-enhanced CT scan shows a fibrous area due to underlying fibrosis, whereas, in the late stage, the fibrotic tissue appears with delayed enhancement [19]. Pure-form GP can be differentiated from the segmental one by Multidetector Computed Tomography (MDCT) imaging. The segmental form of GP shows a focal hypodense lesion in the head of the pancreas proximal to the duodenum, and even more, mild ascending dilatation in the body and tail of the pancreas may appear. On the other hand, in the pure form of the disease, the pancreas appears normal [20]. Furthermore, Magnetic Resonance Imaging (MRI) findings display a characteristic hypointense mass on T1-weighted MR images, and a slightly hypertense on T2-weighted MR images, with delayed contrast enhancement, due to the existing fibrosis. T2-weighted images may also reveal cystic lesions of the duodenal wall and the groove [17,21,22]. Magnetic resonance cholangiopancreatography (MRCP) can detect abnormalities of the main pancreatic duct, the common bile duct, and the ampulla of Vater [6,15,18]. Compared to CT, MRI can visualize the involvement of the pancreas better since the head of the pancreas shows decreased T1 signal intensity due to parenchymal atrophy and fibrosis [23]. Recently, Kalb et al. obtained an accurate diagnosis of 87.2% for groove pancreatitis using 3 MRI criteria: focal thickening of the duodenum, increased contrast agent intake οf the second segment of the duodenum, and cystic alterations in the region of the accessory pancreatic duct, with a negative predictive value for cancer of 92.2% [24]. Finally, Endoscopic Ultrasound (EUS) and Endoscopic Retrograde Cholangiopancreatography (ERCP) can reveal smooth tubular stenosis of the common various bile duct with the main pancreatic duct remaining intact. In addition, EUS represents one of the most sensitive methods for detecting pancreaticobiliary lesions and might detect stenosis of the second part of the duodenum, cysts or pseudocysts of the duodenal wall, and masses located in the groove area or the head of the pancreas, while at the same time providing the possibility of taking biopsies with terms of EUS-Fine Needle Aspiration (FNA) [15,16].

Pancreatic head carcinoma, periampullary cancer, pancreatic groove neuroendocrine tumor, cystic dystrophy of the duodenum, and plastron presence in acute pancreatitis are included in the differential diagnoses of GP [9,18,22]. Differentiating GP from peripancreatic head adenocarcinoma can be very challenging due to the similarities of their clinical and radiological findings [25]. A significant difference is that the vascular involvement found in adenocarcinoma is not observed in GP, while cystic lesions are more commonly found in GP. Additionally, the duodenal wall thickening characterizing GP is not usually present in adenocarcinomas [22].

The treatment options of GP are conservative measures and surgical intervention. As regards the conservative treatment, it aims to substitute the pancreatic function with a pancreatic enzyme supplement and relieve the patient’s pain with sufficient analgesia [25]. Other measures that help pancreatic function recovery are avoidance of smoking and alcohol consumption, parenteral nutrition, and proton pump inhibitors [15,17]. Endoscopic drainage or stenting of the pancreatic duct poses an intervention less invasive than surgery, and it is preferred in patients not suitable for surgery due to their general condition and several comorbidities [26]. In case patients are not clinically improved, present with complications, or the possibility of malignancy cannot be ruled out, the most suitable approach is surgery. Pancreatoduodenectomy, with Whipple’s procedure or with preservation of pylorus–Longmire technique, is the most commonly used method because it provides both a definitive cure of the symptoms and resection of possible malignancy [4,7,10]. Casetti et al. observed a percentage of 76% pain remission in patients after pancreatoduodenectomy [7]. In the event of pyloric or duodenal stenosis, a condition present in our case, procedures such as duodenal or biliary bypass constitute a therapeutic option. Otherwise, in the absence of duodenal stenosis, an alternative surgical approach is groove resection of the pancreatic head (GRPH), which includes the resection of the groove area of the pancreatic head with preservation of the duodenum, common bile duct, main pancreatic duct, and the majority of the pancreatic head [10].

## 4. Conclusions

Groove Pancreatitis is a very rare type of chronic pancreatitis. Patients may present with non-typical symptoms, such as abdominal epigastrium pain, vomiting, and weight loss. Diagnosis of GP poses a challenge to clinicians due to its similarities with pancreatic carcinoma, which sometimes leads to unnecessary over-treatment of patients undergoing extreme interventions that add morbidity or even mortality to the patients. In all cases of pancreatic head abnormalities and duodenal stenosis, suspicion of GP should be raised, primarily in cases where the patient’s history is associated with prolonged alcohol intake. Initial treatment includes conservative measures or less invasive endoscopic techniques. In the failure of the previously mentioned options, surgery is the management of choice. Although, further diagnostic criteria should be established in order to facilitate the early diagnosis and the targeted treatment of patients.

## Figures and Tables

**Figure 1 jpm-13-00644-f001:**
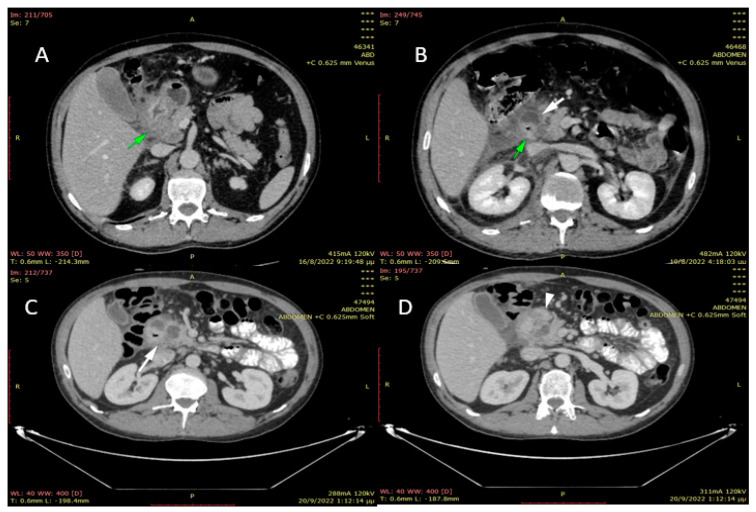
CT findings. (**A**): Admission CT: extensive edema of the duodenal wall and head of the pancreas (arrow). Normal biliary tree. (**B**): 3 days later: duodenal wall edema (green arrow) with cystic lesion formation (white arrow). The adjacent fat tissue is hyperdense due to inflammation. There is also free fluid collection in the anterior pararenal region. (**C**): 1 month after onset of symptoms: subtle increase of the cystic lesions on the duodenal wall. Extensive gastric and duodenal wall edema. (**D**): Same CT: Tissue to gastric antrum that enhances exactly the same as the rest of the pancreas (arrowhead) - ectopic pancreatic tissue Fat stranding and small fluid collection.

**Figure 2 jpm-13-00644-f002:**
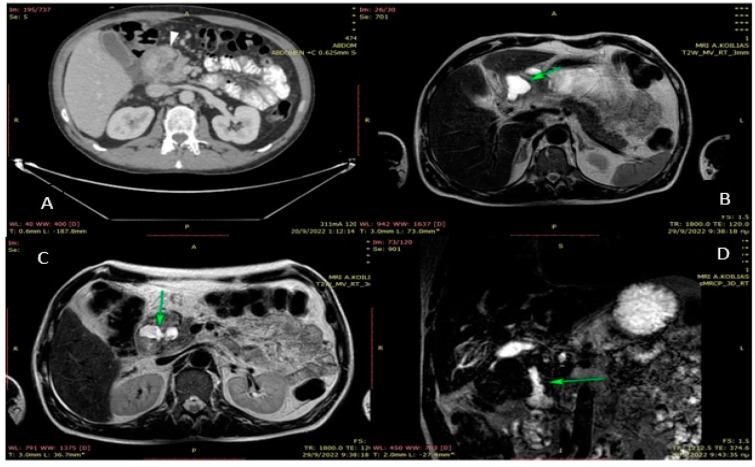
MRI findings. (**A**): Same CT: Tissue to gastric antrum that hances exactly the same as the rest of the pancreas(arrowhead)-ectopic pancreatic tissue Fat stranding and small fluid collection. (**B**): Encapsulated subhepatic fluid collection. (**C**): Cystic lesions to the duodenal wall with proteinic content. Edema to the duodental wall and pancreatic head. (**D**): MRCP: Stenosis of the descending duodenum.

## Data Availability

The data that support the findings of this study are available on request from the corresponding author A.M.

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
