# Peer review of "Surgical Management of Groove Pancreatitis: A Case Report"

_jpm, 2023, doi:10.3390/jpm13040644_

Round 1

Reviewer 1 Report

This case report represented a rare pancreas disorder and it should be read by other physicians. This is useful for doctors to be cautious when a physical examination has been performed in order to ensure that the correct diagnosis has done.

Author Response

We want to thank the Reviewer for the comments. 

Reviewer 2 Report

Abstract

1) Authors say

Differential diagnosis of pancreatic cancer is difficult due to the lack of diagnostic management and the restrictive number of patients

This should be rephrased to a coherent English.

2) Authors say

, to bypass the duodenum by creating gastroenteroanastomosis leading to

This should be rephrased

3) Authors  say

a less morbid procedure can be performed in evidence of malignancy absence

This should be rephrased

Introduction

4) The authors say

is a rare form of chronic disease

It is quiet usual to find this mantra repeated in many publications on the subject.

It is also time to finish with this mantra that has no real fundamentals in statistics:

In about 20% of patients undergoing pancreaticoduodenectomy to treat chronic pancreatitis, groove pancreatitis is detected.

Tezuka K, Makino T, Hirai I, Kimura W. Groove pancreatitis. Dig Surg. 2010;27(2):149-52. doi: 10.1159/000289099. Epub 2010 Jun 10. PMID: 20551662.

This reference is among the references in the paper. Therefore I would suggest the authors to read their own references.

5) Authors say

which is not clearly determined yet

Correct syntax

6) authors say

GP usually affects 40-50-year-old alcoholic male patients and is presented with epigastric

Correct syntax and rephrase.

7) Authors say

Based on those clinical and radiological findings, the suspected diagnosis was Groove pancreatitis o…

Groove pancreatitis is a segmental chronic pancreatitis that affects the anatomical area between the pancreatic head, the duodenum, and the common bile duct, referred to as the groove area. Therefore "groove" is an anatomical structure and not the proper name of a researcher. Therefore it should not be written with a capital letter

8) Authors say

several days later on,

Correct syntax

Discussion

9) The authors repeat the mantra of  groove pancreatitis as a rare form of chronic pancreatitis. Well, it is not rare.

10) Mellitus should not be written with a capital M

11) several methods are being used through the years

Correct syntax

12) MDCT imaging

I do not know what MDCT image is. Please write the full name of the study before starting with abbreviations. Readers, as ignorant as I am, do not like to guess what the authors are including in their abbreviations.  

13) Compared to CT, MRI can visualize in a better way the involvement of the pancreas since the head of the pancreas showed decreased T1 signal intensity due to parenchymal atrophy and fibrosis

Please pay attention to the verb tense.

14) hyper uptake of the second segment of the duodenum,

Please convert this into English.

15) EUS

I do not know what EUS is, and possibly many readers will have the same problem.

16)  ERCP

I happen to know what ERCP means but it would be easier for the readers if the authors write the whole name of the procedure.

Author Response

Authors’ answer to Reviewers’ comments

REVIEWER 2

First of all, we appreciate all the reviewer’s comments. We made all the changes that were proposed.

  • At the comments 1-2-3-5-6-7-8-10-11-13: the text was rephrased and corrected as advised
  • At the comments 12-15-16, the terms MDCT, EUS, and ERCP were clarified and analyzed
  • For comments 4 & 9: It should be noted that the term ‘rare’ has been removed. Nevertheless, we are aware of the final incidence rates of this disease. However, many physicians are not familiar with this entity and, in particular, the term was used because its diagnosis before the histological confirmation is difficult. As a result, the number of patients undergoing major surgical interventions with increased morbidity and mortality, with preoperative suspicion of malignancy, is not limited. We acknowledge that it should have been better clarified.